# The Impact of the Age, Dyspnoea, and Airflow Obstruction (ADO) Index on the Medical Burden of Chronic Obstructive Pulmonary Disease (COPD)

**DOI:** 10.3390/jcm11071893

**Published:** 2022-03-29

**Authors:** Chin-Ling Li, Mei-Hsin Lin, Yuh-Chyn Tsai, Ching-Wan Tseng, Chia-Ling Chang, Lien-Shi Shen, Ho-Chang Kuo, Shih-Feng Liu

**Affiliations:** 1Department of Respiratory Therapy, Kaohsiung Chang Gung Memorial Hospital, Kaohsiung City 833, Taiwan; musquito16@cgmh.org.tw (C.-L.L.); n3023@cgmh.org.tw (M.-H.L.); jane2793@cgmh.org.tw (Y.-C.T.); wanda2640@cgmh.org.tw (C.-W.T.); ben64@cgmh.org.tw (C.-L.C.); nancy@cgmh.org.tw (L.-S.S.); erickuo48@yahoo.com.tw (H.-C.K.); 2College of Medicine, Chang Gung University, Taoyuan 333, Taiwan; 3Department of Pediatrics, Kaohsiung Chang Gung Memorial Hospital, Kaohsiung City 833, Taiwan; 4Division of Pulmonary and Critical Care Medicine, Department of Internal Medicine, Kaohsiung Chang Gung Memorial Hospital, Kaohsiung City 833, Taiwan

**Keywords:** chronic obstructive pulmonary disease, ADO index, Charlson comorbidity index, medical burden

## Abstract

There are currently no good indicators that can be used to predict the medical expenses of chronic obstructive pulmonary disease (COPD). This was a retrospective study that focused on the correlation between the age, dyspnoea, and airflow obstruction (ADO) index and the Charlson comorbidity index (CCI) on the medical burden in COPD patients, specifically, those of patients with complete ADO index and CCI data in our hospital from January 2015 to December 2016. Of the 396 patients with COPD who met the inclusion criteria, 382 (96.5%) were male, with an average age of 71.3 ± 8.4 years. Healthcare resource utilisation was positively correlated with the ADO index. A significant association was found between the ADO index and CCI of COPD patients (*p* < 0.001). In-hospitalization expenses were positively correlated with the CCI (*p* < 0.001). Under the same CCI, the higher the ADO score, the higher the hospitalisation expenses. The ADO quartiles were positively correlated with the number of hospitalisations (*p* < 0.001), hospitalisation days (*p* < 0.001), hospitalisation expenses (*p* = 0.03), and total medical expenses (*p* = 0.037). Findings from this study show that the ADO index can predict the medical burden of COPD.

## 1. Introduction

Chronic obstructive pulmonary disease (COPD) is a common chronic respiratory disease characterised by progressive disease and irreversible airflow limitation and is the third leading cause of death globally [1]. As of 2021, an estimated 328 million people have COPD worldwide [2].

COPD is a complex disease. In addition to lung inflammation, it also has the characteristics of systemic inflammation [3] and often coexists with other diseases. Complications are common in COPD patients at any stage of the disease [4]. Approximately 73% of COPD patients suffer from one or more comorbidities [5], and multiple comorbidities are common among COPD patients.

The Charlson comorbidity index (CCI) was first developed by Charlson et al. A higher CCI indicates a greater number and severity of comorbidity, with higher medical needs that lead to a higher medical burden [6,7]. COPD patients with multiple comorbidities were 4.7 times more expensive than those without comorbidities [8]. In COPD patients, the progression of the disease to death is slow, which worsens with the severity of the disease and age [9]. Disease severity of COPD is a considerable economic burden for healthcare [10]. Comorbidities are also the important factors that determine COPD patients’ medical expenses and prognosis [11].

Because COPD is a complex disease, unable to be predicted by FEV1 alone, many multi-dimensional assessment tools are currently used to evaluate COPD disease severity, acute exacerbation, and are prognosis, such as COTE COPD-specific comorbidity test (COTE); dyspnoea, obstruction, smoking, and exacerbation (DOSE) index; body mass index, obstruction, dyspnoea, exercise (BODE index); and age, dyspnoea, airflow obstruction (ADO) index. In 2009, Puhan et al. developed a simpler ADO index to predict the three-year mortality of patients with moderate to severe COPD [12]. ADO index was also thought of as the best predictor for COPD risk assessment in a recent study [13]. Moreover, reviews suggest that the ADO score is the most discriminatory prognostic score for predicting mortality among patients with COPD [14]. Additionally, the ADO index is easier to be obtained than other indices, and it is more convenient to use in clinical practice. However, ADO index predicting medical burden for COPD is unknown. This study aimed to analyse the correlation between the ADO index, CCI, and healthcare resource utilisation in COPD.

## 2. Materials and Methods

### 2.1. Study Design 

ADO has proved to be the best index in assessing the risk of COPD [13], however, the value of ADO in assessing medical expenses is still unknown. This retrospective study used the patient database of our previous study [15] to explore the correlation between ADO and medical expenses. There are complete ADO, CCI, and medical expenses such as hospitalization and emergency treatment of patients in the database.

### 2.2. Study Population

A flow chart of the study is shown in Figure 1, which is a retrospective study of clinical outcomes. The data were drawn from the electronic database of the Kaohsiung Chang Gung Medical Center (KCGMH) and were reviewed and approved by the Institutional Review Board Committee (IRB: 201701293B0) of the Chang Gung Medical Center. The IRB of the Chang Gung Medical Center did not require patients to agree to a review of their medical records for this retrospective study. This study complied with the Declaration of Helsinki and Good Clinical Practice Guidelines and was approved by the Ethics Committee of the Chang Gung Medical Center.

The patients’ retrospective reports of the 6-min walking test were dated from 31 January 2015 to 31 August 2017 (a total of 32 months) *n* = 1063.

The inclusion criteria of patients with COPD in KCGMH were as follows: (1) diagnostic code ICD-9-CM: 490–496, ICD-10-CM: J41-J44; (2) diagnosed as COPD by a post-bronchodilator FEV1/FVC < 70% with expiratory flow obstruction; (3) those who had complete clinical record data, BODE index, ADO index, CCI and medical cost data. The exclusion criteria include as follows: (1) those who were less than 40 years old, (2) those whose lung function test did not match with COPD diagnostic criteria according to the COPD GOLD guideline [16], (3) incomplete clinical record data. 

### 2.3. Clinical Variables 

The ADO index of COPD patients was categorised into four quartiles, same as that for the BODE index, as follows: quartile one, 0–2; quartile two, 3–4; quartile three, 5–6; and quartile four, 7–10 [17,18]. CCI was categorised into two groups: the high CCI ≥ 3 group and low CCI < 3 group [19,20]. Variables included age, gender, pack-years, pulmonary function test, modified Medical Research Council dyspnoea scale (mMRC), body mass index (BMI), DLCO, 6MWD, BODE index, ADO index and CCI, the number of outpatient visits, the number and days of hospitalization, and medical expenses.

### 2.4. Statistical Analysis 

Baseline characteristics are expressed as the mean and standard deviation (mean ± SD), median (interquartile range, IQR), or *N* (%). The distributions of ADO and CCI were evaluated by descriptive statistical analysis. ADO quartiles and two CCI groups by Chi-square tests, Fisher’s exact test, and one-way analysis of variance (one-way ANOVA) were applied to compare the differences in variables by quartiles, and then posteriori comparisons were made using Scheffé’s test. Lastly, one-way ANOVA and linear contrasts were applied to check for linear trends. Data analysis was performed using the IBM Statistical Package for the Social Sciences (SPSS) version 26.

## 3. Results

The cohort consisted of 396 patients with COPD. In the study, we estimated the original ADO index score in our COPD patients based on the cut-off values of FEV1 (% predicted), dyspnea (mMRC scale) and age (years) in a previously published report by Puhan et al [12] (Table 1). Table 2 shows the baseline characteristics of the study population. The mean age was 71.3 years ± 8.4 years, of whom 96.5% were male. In terms of smoking status, the average length of smoking was 31.7 years ± 18.5 years. The average body mass index (BMI, kg/m^2^) was 23.5 ± 4.1.

According to the definition of severity of COPD in the GOLD guidelines [21], 82.6% of patients had moderate to severe COPD, the largest subset in this study. Among them, 187 patients had moderate COPD, making up 47.2% of the total, and 140 patients had severe COPD, making up 35.4% of the total. The CCI had a mean score of 3.3 ± 2.8; mMRC, 1.72 ± 0.9; the BODE index, 3.3 ± 2.1; and ADO index, 4.9 ± 1.8 (quartile one: 10.1% [*n* = 40]; quartile two: 31.3% [*n* = 124]; quartile three: 38.4% [*n* = 152]; quartile four: 20.2% [*n* = 80]). The mean distance of the 6-MWD was 351.9 ± 11.6 m (Table 2).

Figure 2 shows a significant correlation between the ADO quartiles and CCI (*p* < 0.001).

In addition, CCI was divided into two groups: high (CCI ≥ 3) and low (CCI < 3) levels. The hospitalization expenses by the ADO quartiles for the high CCI and low CCI groups. Patients with a high CCI level had higher hospitalization expenses than those with a low CCI level after adjusting for the ADO quartiles (*p* < 0.001). In the high CCI group, compared to quartile 1 hospitalization expenses NT$ 6562; quartile 2 hospitalization expenses, NT$ 57,601; quartile 3 hospitalization expenses, NT$ 70,488; and quartile 4 hospitalization expenses, NT$ 62,038 showed an increased trend. In the low-CCI group, the hospitalization expense for quartile 1 was NT$ 12,624; quartile 2, NT$ 14,930; quartile 3, NT$ 20,954; and quartile 4, NT$ 2615 showed no statistical significance (*p* = 0.651).

In these patients, the average number of outpatient visits and outpatient medical expenses were not significantly affected by the ADO quartiles (*p* = 0.108) (Table 3). Inpatient medical service needs were divided into the number of hospitalisations and the number of days of hospitalisation. The ADO quartile was positively linearly related to the number of hospitalizations (*p* < 0.001). The results of the post hoc comparison with Scheffé’s method are as follows: The number of hospitalisations of quartile four was 1.34 times more than that of quartile one (*p* < 0.001) and the number of hospitalisations of quartile four was 1.13 times more than that of quartile two (*p* < 0.001). 

ADO quartiles were positively and linearly associated with hospitalization costs, and with increasing ADO quartile levels, hospitalization costs also increased (*p* = 0.03) (Table 3). A linear trend of the length of hospital stay by the ADO quartile (*p* < 0.001). The relation among the ADO quartiles and the length of hospital stay after post hoc comparison with Scheffeé’s method was as follows: The number of hospitalisation days of quartile four was 17.5 d more than that of quartile one (*p* = 0.001), and the number of hospitalisation days of quartile four was 15.69 d more than that of quartile two (*p* < 0.001), quartile three stayed 7.59 d longer than that of quartile two (*p* = 0.047), a statistically significant difference. Figure 3 shows a linear trend among the ADO quartiles and the total medical expenses (*p* = 0.037): (Table 3) with an increase of ADO quartile level, the total medical expenses also increased.

## 4. Discussion

This study shows that the ADO index correlates with CCI index, and medical burden including hospitalisation frequencies, hospitalisation days, hospitalisation expenses, and total medical expenses. In addition to being a predictor for COPD risk assessment, the ADO index also has a clinical value in predicting healthcare costs.

COPD is a complex disease. With the increase of age, there are more comorbidities, especially the comorbidities of the heart and lung, and lung function also declines year by year, so the degree of dyspnoea is also higher. Although the components of the ADO index are free of comorbidities, it is easy to understand that ADO and CCI are highly correlated. However, regardless of the low CCI or the high CCI group, ADO has a positive correlation trend with medical expenses, especially in the high CCI group, which means that using combined ADO and CCI can more accurately estimate medical expenses.

COPD imposes substantial costs on health systems, mainly associated with moderate-to-severe stages and consequent exacerbations and complications [22]. Although ADO was not previously reported in the literature to be associated with medical costs, the components of ADO including age, dyspnoea, and airflow obstruction are individually related to COPD exacerbations. Comorbidities increase and lung function decreases with age in COPD patients [23], which is related to future exacerbations and related to complications. Additionally, research analysis indicates that patients with a CCI score of 3 or above are equivalent to two chronic diseases or one serious disease [7,24]. Older people usually have higher comorbidities and higher medical expenses [25]. Breathlessness is one of the most frequent symptoms in COPD, which may result in disability, decreased productivity, increased comorbidities, and increased healthcare costs [26]. Patient-reported symptoms also provide important information about future COPD exacerbations and exacerbation-related costs [27]. Studies have shown that COPD patients with higher mMRC scores have higher emergency room and hospitalization rates, and that mMRC ≥ grade two in COPD patients is associated with higher healthcare and social costs [26]. Pulmonary function testing is a tool for diagnosing the degree of airflow obstruction in COPD [28], but FEV1 is not sufficient to measure the overall severity of COPD. However, FEV1 remains a useful reference for physicians to assess clinical severity or treatment. In particular, decreased FEV1 was associated with exacerbation frequency [28].

Both BODE and ADO indices can be used to predict the prognosis of COPD patients, but the ADO index data is easily available, widely used, and has similar predictive power to the revised BODE index [29]. Previous studies have found that the BODE index is related to medical resources [15,30,31]. Our present research found that the ADO index correlates with medical costs. However, whether BODE or ADO is better for predicting medical resources COPD requires further research.

According to our data, patients with ADO quartile four scores had fewer outpatient medical needs than those in quartile three. We speculate that the reasons could be as follows: (1) patients with ADO quartile four have high disease severity, which may cause a higher frequency of hospitalisation and longer hospitalisation stays, which leads to a decreased number of outpatient visits; (2) the age of ADO quartile four was older than that of quartile three, and the average symptoms of dyspnoea and reduced mobility of patients with were higher than that of ADO quartile three. Patients with ADO quartile four may increase the difficulty of visiting outpatient clinics, as they need more assistive devices such as wheelchairs, oxygen, and caregiver assistance to go out to fulfill their medical needs, thus leading to a lower outpatient clinical visit rate than in patients with ADO quartile three.

There were some limitations to our study. First, as this was a retrospective study, a prospective study is needed to prove the clinical outcomes. Our recent literature review found that there are prospective studies on ADO and COPD, all discussing the predictive power of mortality or disease risk assessment [13,32], and almost no literature discusses ADO and medical burden. Second, almost all of the patients are male. Gender may modify and affect outcomes, disease management, and social health costs. The original database of this study was collected from COPD patients who had received 6 MWT [15]. It is a fact that most of our enrolled COPD patients during that period were male, and we did not specifically exclude females. More than 80% of the COPD patients in Taiwan are men [33,34] and it may be that the number of women who smoke in Taiwan is far less than in Western countries. In addition, the source of the case data also came from a single medical centre, which may not be more objective and not representative of all COPD patients. Third, medical expenses are divided into direct medical expenses and indirect medical expenses. The data we collected are direct medical expenses. Our study lacked data on indirect medical costs associated with COPD and comorbidities. These indirect medical costs include time spent in doctor visits, lost productivity, lost productivity due to early retirement, disability pension payments, and care costs. Fourth, medical costs may vary with each doctor’s treatment. In 1995, Taiwan launched a single-payer National Health Insurance (NHI) program. As of 2014, over 99% of Taiwan’s population was enrolled. Taiwan Bureau of National Health Insurance has started to use the diagnosis-related group system (Tw-DRG) since January 2010. There is a unified payment standard for medical expenses. The hospitals will then claim the related charges from the government [35]. Health Insurance Bureau will pay reasonable medical expenses (not always pay 100%) to the hospital according to the DRG disease review system. Moreover, our hospital is a large medical center in Taiwan and implements the attending physician system. The attending physicians are all professionally trained pulmonologists. These COPD patients are all patients of the attending pulmonologist. The treatment of the patient by the attending physician should be generally consistent.

## 5. Conclusions

The ADO index is a convenient and effective clinical evaluation tool for COPD. This study demonstrates that the ADO index is correlated with the number of hospitalisations, hospitalisation days, hospitalisation expenses, and total medical expenses. Therefore, the ADO index has a value in predicting the medical burden for COPD.

## Figures and Tables

**Figure 1 jcm-11-01893-f001:**
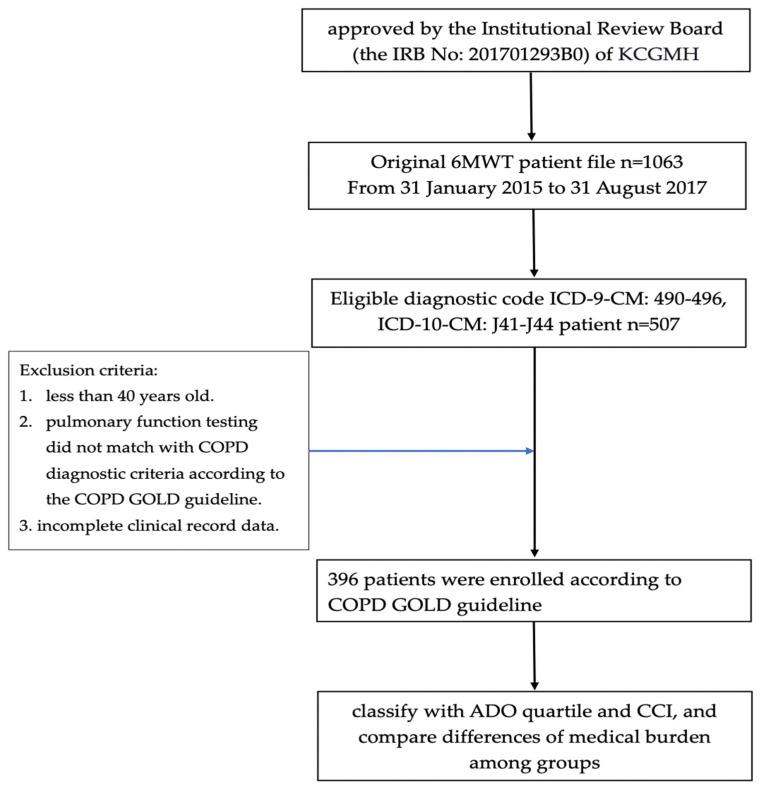
Flow chart of selected participants in this study.

**Figure 2 jcm-11-01893-f002:**
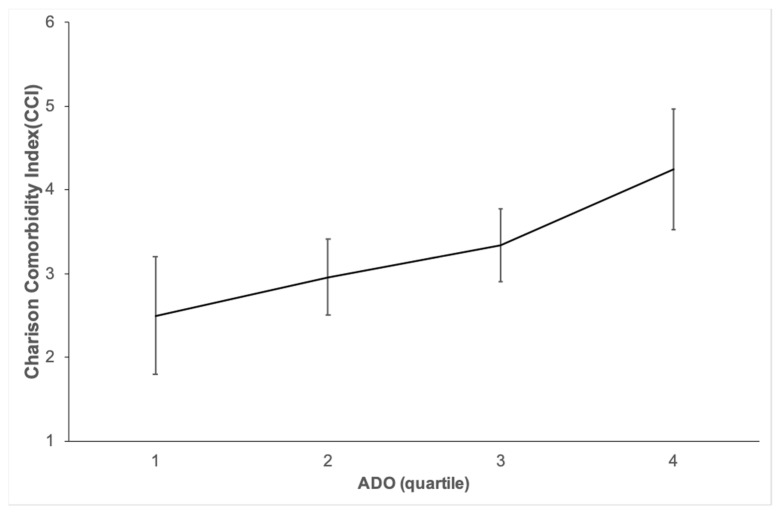
This shows the good correlation between the ADO quartiles and the Charlson comorbidity index (CCI) (*p* < 0.001).

**Figure 3 jcm-11-01893-f003:**
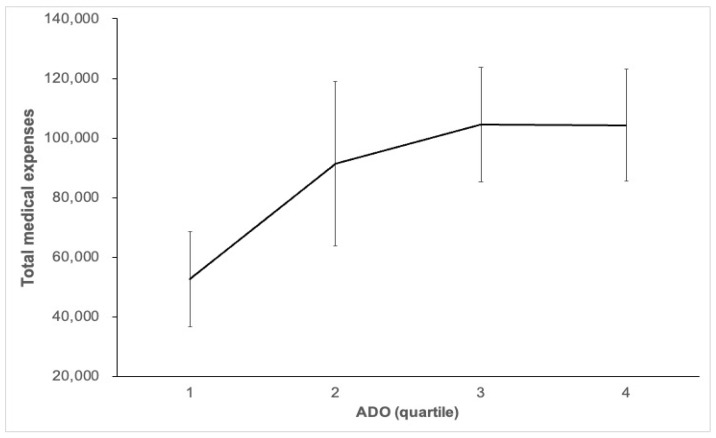
This shows a linear trend between the ADO quartiles and total medical expenses (*p* = 0.037).

**Table 1 jcm-11-01893-t001:** Assignment of points for the age, dyspnoea, airflow obstruction (ADO) index.

Points	0	1	2	3	4	5
FEV1 (% predicted)	≥65%	≥36–64%	≤35%			
Dyspnoea (MRC scale)	0–1	2	3	4		
Age (years)	40–49	50–59	60–69	70–79	80–89	≥90

FEV1 = forced expiratory volume in 1 s; MRC = Medical Research Council. Note: adapted from [12], Copyright on 11 December 2021.

**Table 2 jcm-11-01893-t002:** Baseline characteristics of enrolled 396 chronic obstructive pulmonary disease (COPD) patients.

Factors	Mean ± Standard Deviation (SD) or *n* (%)
Male (%)	382 (96.5)
Smoking history (pack-years)	31.7 ± 18.5
Age (years)	73.1 ± 9.5
Body-mass index (BMI)	23.5 ± 4.1
FVC (% of predicted value)	79.7±16.7
FEV1/FVC (%)	52.7 ± 10.6
FEV1 (% of predicted value)	55.2 ± 18.2
GOLD stage (%)	
Mild	46 (11.6)
Moderate	187 (47.2)
Severe	140 (35.4)
Very severe	23 (5.8)
DLCO (%)	68.5 ± 21.0
6-MWD (m)	351.9 ± 111.6
mMRC	1.72 ± 0.9
mMRC dyspnea scale	
Scale 0/1/2/3/4	25/133/173/56/9
CCI	3.3 ± 2.8
BODE INDEX	3.0 ± 2.1
ADO INDEX	4.9 ± 1.8
ADO quartile: Q1, Q2, Q3, Q4	(%) *
quartile one	40 (10.1)
quartile two	124(31.3)
quartile three	152 (38.4)
quartile four	80 (20.2)

* Quartile one was defined by a score of 0–2, quartile two by a score of 3–4, quartile three by a score of 5–6, and quartile four by a score of 7–10. Abbreviations: CCI, Charlson comorbidity index; ADO index, composite index of age, dyspnoea, and airflow obstruction; BODE index, composite index of body mass index, airflow maximum expiratory pressure obstruction, dyspnoea, and exercise capacity; 6 MWD, 6-min walking distance; FEV1, forced expiratory volume in 1 s; FVC, forced vital capacity; MRC score, Medical Research Council dyspnoea scale.

**Table 3 jcm-11-01893-t003:** Value of medical burden and ADO quartile.

Classification	ADO Quartile	Frequency or Costs (Mean (95% CI))	*p*-Value
number of outpatient visits	1	14.68 (11.92–17.43)	0.108
	2	17.3 (15.26–19.33)	
	3	22.68 (17.64–27.73)	
	4	19.08 (16.55–21.6)	
outpatient medical expenses	1	40,843.48 (30,640.1–51,046.9)	0.344
	2	65,834.9 (44,039.02–87,630.877)	
	3	69,271.15 (54,892.98–83,649.32)	
	4	62,963.4 (54,702.35–71,224.45)	
number of hospitalizations	1	0.35 (0.11–0.59)	<0.001
	2	0.56 (0.38–0.73)	
	3	1.11 (0.82–1.4)	
	4	1.69 (1.11–2.26)	
Length of hospital stay	1	2.75 (0.53–4.97)	<0.001
	2	4.56 (2.74–6.39)	
	3	12.16 (8.37–15.94)	
	4	20.25 (12.6–27.9)	
hospitalization expenses	1	11,865.83 (−859.49–24,591.14)	0.03
	2	25,597.92 (9280.94–41,914.9)	
	3	35,292.59 (23,548.79–47,036.39)	
	4	41,407.2 (24,540–58,274.4)	
total medical expenses	1	52,709.3 (36,714.96–68,703.64)	0.037
	2	91,432.81(63,742.03–119,123.6)	
	3	104,563.74(85,295.18–123,832.31)	
	4	104,370.63 (85,685.52–123,055.73)	

## Data Availability

The data supporting this research is available from S.-F.L. and C.-L.L.

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
