# Peer review of "The Impact of the Age, Dyspnoea, and Airflow Obstruction (ADO) Index on the Medical Burden of Chronic Obstructive Pulmonary Disease (COPD)"

_jcm, 2022, doi:10.3390/jcm11071893_

Round 1

Reviewer 1 Report

As you mention in discussion for limitiation for the study : "First, as this was a retrospective study, a prospective study is needed to prove the clinical outcomes."-It will be better to compare retrospective and prospective data.

Restrospective data are form 2015 and 2016 , and the data will publish in 2022. It is too much time gap.

Lines 198-202 – used literatures are too old for discussion (1998, 2011 x 2) for comparing the results from the study

Lines 204-208 – used only one literature for comparing (2009), nothing newly

Line 212 – „In the literature review, there are few studies related to ADO indicators and medical resource utilisation“ – Those studies are not in literature; you have to write the numbers.

Literature number 1 is without year of publishing

Reviewer 2 Report

The work presented by the authors is quite original and highlights the importance of using tools to assess the severity of COPD, the clinical and progressive deterioration of the disease in order to highlight the importance of monitoring and managing these patients. . The aim of this original paper is to analyze the correlation between the ADO index, CCI, and healthcare resource in patients with COPD.

Introduction:
The introduction is well written and concise. Current and recent references are cited.

Materials and methods
The patient selection flow-chart is well structured and evaluates the whole screening process well.

Statistical methodology
The statistical methods used are exhaustive

Results
Well written and clear to the readers.

Discussion
Well written. Improve native English.
In the discussion discuss more about the limitation that the authors themselves highlight. Almost all of the patients are men. This could influence some of the results of the tools considered.
In fact, sex modifies and influences the prognosis, the management of the disease but also the socio-health costs.
Why was only the male population selected? Or rather, what could a majority of men have determined in the period of retrospective observation? Authors should better emphasize this limitation and give an explanation if necessary.
Even if there are limitations, as described by the authors, the work is still interesting and can be useful for the scientific community.

Reviewer 3 Report

The study aimed to analyse the correlation between the airflow obstruction index, the Charlson comorbidity index, and healthcare resource utilisation in chronic obstructive pulmonary disease.

Is a retrospective study that use data of the medical records and medical burden from the Medical Center, a dataset for administrative purposes with intrinsic limits for clinical evaluations.

The manuscript requires substantial clarifications. In particular:

The introduction is too long and should be shortened, much of what is reported is well known in the literature. Focus on what is necessary to motivate the study and qualify the difference from what already exists in the literature.

The limitations of the study, which you clearly describe, are too important to justify the results obtained.

Materials and methods

Data source: the codes used must be illustrated because they are probably specific of your database and therefore not common in other contexts.

How did you define the resources used for individual cases? Have you derived the numbers from the total consumption of the department? If so, there would be too many biases to justify the expenses incurred. The use of resources must be specified because the assumptions you make are too generic.

Study population: there is no definition and evaluation of the appropriateness of the treatments applied. How did you manage to differentiate the costs for tests and treatments that were not useful or were accidentally and incorrectly used?

How could you explain that the cases were almost all male during the period covered by the data collection?

What kind of randomization of the patients you have done? It is necessary to clarify in order to avoid bias.

You have described the inclusion criteria but you have not reported the exclusion criteria, these must also be clearly specified.

Results

In table 1 you refer to the score assigned to dyspnea. How did you manage to differentiate between pulmonary and cardiac dyspnea?

Pag 3 line 120 you refer to “definition of severity of COPD in the Gold guidelines”, please include the reference.

Page 3 line 121 you refer to "82.6% of patients had moderate to severe COPD". It is necessary to analyze the ADO and the CCI for the two individual groups (moderate and severe) clearly because the total support they receive is completely different.

Table 2 should be summarized, it contains some superfluous data that can cause confusion. For treated patients, please consider what has been mentioned in the previous paragraph.

Pag 5 line 143 you refer to “In addition, CCI was divided into two groups…” this must be clearly referred to the individual “moderate and severe” groups.

Figures 3 and 4 can be deleted because they do not add anything to what has already been said in the text.

Also figures 5, 6 and 7 are redundant. Not having a complete division in patients with moderate pathology (how did you define and classify it? Which method did all the operators who compiled the folder used? final?) the data you report are not informative about the real costs of your patients.

Page 6 line 161 reference is made to "The needs of the hospital medical service have been divided into number of hospitalizations and number of days of hospitalization". How did you determine if hospitalization and its duration were appropriate? Many conditions can affect hospitalization and its duration without being included in the real costs of a specific treatment. How did you manage to highlight all the incorrect hospitalizations and the total treatments applied?

The discussion must be reviewed once the above has been clarified and specified. It is necessary to avoid repetition of the results you have already described. Essentially it is necessary to compare and comment your results with those existing in the literature, highlighting why your study is important.

Some bibliographic entries are outdated and some of what has been reported has been superseded by more recent literature.

Reviewer 4 Report

Thank you for your paper, I readi t with great interest. I think it has clinical relevance but the quality of the manuscript needs to be improved with further justification and clarity. I have provided the below suggestions in the spirit of support. I hope you find these of benefit.

In the introduction you clearly state the burden of COPD and reasons why comorbidities are so important to consider in prognosis, perhaps add some data re length of stays for those with vs without high comorbid disease burden. If these data are not available this should be clearly stated as highlights the gap further.

Good info re range of prognostic outcome measures is provided.

I think you have made an error stating in line 60: “it is easier to obtain the 6-min walking distance (6MWD) with age” instead of stating ADO.

What is your hypothesis and null hypothesis?

Justify why the data are 5 years old

Methods:

Inclusion criteria states less than 40 years old. Do you mean over 40 years old and why was this the predetermined age?

Give further description of medical expenses that were considered and did you use a validated health economic questionnaire for this?

Your aim was to look for correlations between ADO and CCI, be clear which statistics were used for this aim and why Pearson or Spearman correlations were not considered.

Results:

A bit of a red flag here is that 96.5% were male. Why don’t women with COPD get hospitalised? Detail and critique the selection bias here.

Provide the correlation test statistic as well as the p value, with the reference ranges for poor, good, very good correlation in the methods prior to this.

Justify with a reference perhaps your dichotomisation of high and low CCI groups

Be clearer whether hospital expenses relates to length of stay or a combination of factors.

I don’t think you need to present the Figure 6 associated data for ADO hospitalisation expenses if your CCI hospitalisation expenses are adjusted for ADO.

I think you need to be more specific presenting the data that relate to your primary hypothesis in your results and perhaps include other results in the appendices otherwise this risks losing clarity of your message and potential lack of focus.

Discussion

Very limited discussion about the important findings of the CCI as a predictor adjusted of AO in your discussion.

Think you need brief statements at the start of your discussion on the novelties of your work.

Good point made about retrospective nature and difference between indirect costs and direct costs.

You need to discuss the limitation of huge majority being male with very limited external validity for women.

What do the findings of greater CCI and ADO mean in clinical practice? What can clinicians do differently as a result of your work. What specific research questions and associated study designs are needed as a result of your work?

Round 2

Reviewer 1 Report

Thank you for changes, the study paper is ok now

Reviewer 4 Report

Dear Authors,

Thank you for your revised manuscript. It is clear you have made significant changes to all sections in response to initial reviewer comments. However, the added text to the manuscript in parts needs revision to improve the english language used. Please revise the following statements:

"Although ADO did not be reported to associated with medical cost before"

"ADO index predicting medical burden for COPD was not reported before. Our research found that the ADO index correlates with medical costs. Which is better BODE or ADO to predict healthcare resource needs further research"

"Fourth, is there a difference in medical treatment that causes a gap in medical expenses?" - this should not be written as a question.

"Most pulmonologist treat the similar way, with minor differences."

"The ADO index is a convenient and effective clinical evaluation tool for COPD with obvious advantages.” - delete with obvious advantages.
